# Outcome Prediction Based on Automatically Extracted Infarct Core Image Features in Patients with Acute Ischemic Stroke

**DOI:** 10.3390/diagnostics12081786

**Published:** 2022-07-23

**Authors:** Manon L. Tolhuisen, Jan W. Hoving, Miou S. Koopman, Manon Kappelhof, Henk van Voorst, Agnetha E. Bruggeman, Adam M. Demchuck, Diederik W. J. Dippel, Bart J. Emmer, Serge Bracard, Francis Guillemin, Robert J. van Oostenbrugge, Peter J. Mitchell, Wim H. van Zwam, Michael D. Hill, Yvo B. W. E. M. Roos, Tudor G. Jovin, Olvert A. Berkhemer, Bruce C. V. Campbell, Jeffrey Saver, Phil White, Keith W. Muir, Mayank Goyal, Henk A. Marquering, Charles B. Majoie, Matthan W. A. Caan

**Affiliations:** 1Department of Biomedical Engineering and Physics, Amsterdam UMC, Location AMC, 1105 AZ Amsterdam, The Netherlands; h.vanvoorst@amsterdamumc.nl (H.v.V.); h.a.marquering@amsterdamumc.nl (H.A.M.); 2Department of Radiology and Nuclear Medicine, Amsterdam UMC, Location AMC, 1105 AZ Amsterdam, The Netherlands; j.w.hoving@amsterdamumc.nl (J.W.H.); m.s.koopman@amsterdamumc.nl (M.S.K.); m.kappelhof@amsterdamumc.nl (M.K.); a.e.bruggeman@amsterdamumc.nl (A.E.B.); b.j.emmer@amsterdamumc.nl (B.J.E.); o.a.berkhemer@amsterdamumc.nl (O.A.B.); c.b.majoie@amsterdamumc.nl (C.B.M.); 3Department of Clinical Neurosciences and Radiology, Hotchkiss Brain Institute, Cumming School of Medicine, University of Calgary, Calgary, AB T2N 1N4, Canada; ademchuck@ucalgary.ca; 4Department of Neurology, Erasmus MC University Medical Center, 3015 GD Rotterdam, The Netherlands; d.dippel@erasmusmc.nl; 5Department of Diagnostic and Interventional Neuroradiology, IADI, Inserm, CHRU, Université de Lorraine, 54500 Nancy, France; s.bracard@chru-nancy.fr; 6CIC-Epidémiologie Clinique, 1433, CHRU, Inserm, Université de Lorraine, 54500 Nancy, France; francis.guillemin@chru-nancy.fr; 7Department of Neurology, Maastricht UMC, 6229 HX Maastricht, The Netherlands; r.van.oostenbrugge@mumc.nl; 8Cardiovascular Research Institute Maastricht, 6229 ER Maastricht, The Netherlands; 9Department of Neurology, Royal Melbourne Hospital, Parkville, VIC 3050, Australia; peter.mitchell@mh.org.au (P.J.M.); bruce.campbell@mh.org.au (B.C.V.C.); 10Department of Radiology, Maastricht UMC, 6229 HX Maastricht, The Netherlands; w.van.zwam@mumc.nl; 11Department of Clinical Neurosciences, University of Calgary, Calgary, AB T2N 1N4, Canada; michael.hill@ucalgary.ca (M.D.H.); mgoyal@ucalgary.ca (M.G.); 12Department of Neurology, Amsterdam UMC, Location AMC, 1105 AZ Amsterdam, The Netherlands; y.b.roos@amsterdamumc.nl; 13Department of Neurology, Stroke Institute, University of Pittsburgh Medical Center, Pittsburgh, PA 15213, USA; jovin-tudor@cooperhealth.edu; 14Department of Radiology and Nuclear Medicine, Erasmus MC University Medical Center, 3015 GD Rotterdam, The Netherlands; 15Department of Medicine, University of Melbourne, Parkville, VIC 3010, Australia; 16Department of Neurology and Comprehensive Stroke Center, David Geffen School of Medicine, University of California, Los Angeles (UCLA), Los Angeles, CA 90095, USA; jsaver@mednet.ucla.edu; 17Translational and Clinical Research Institute, Faculty of Medical Sciences, Newcastle University, Newcastle upon Tyne NE1 7RU, UK; phil.white@newcastle.ac.uk; 18Department of Neuroradiology, Newcastle upon Tyne Hospitals, Newcastle upon Tyne NE1 7RU, UK; 19Institute of Neuroscience and Psychology, University of Glasgow, University Avenue, Glasgow G12 8QQ, UK; keith.muir@glasgow.ac.uk

**Keywords:** acute ischemic stroke, functional independence, follow-up DWI, infarct core image features, infarct core segmentation, support vector machine

## Abstract

Infarct volume (FIV) on follow-up diffusion-weighted imaging (FU-DWI) is only moderately associated with functional outcome in acute ischemic stroke patients. However, FU-DWI may contain other imaging biomarkers that could aid in improving outcome prediction models for acute ischemic stroke. We included FU-DWI data from the HERMES, ISLES, and MR CLEAN-NO IV databases. Lesions were segmented using a deep learning model trained on the HERMES and ISLES datasets. We assessed the performance of three classifiers in predicting functional independence for the MR CLEAN-NO IV trial cohort based on: (1) FIV alone, (2) the most important features obtained from a trained convolutional autoencoder (CAE), and (3) radiomics. Furthermore, we investigated feature importance in the radiomic-feature-based model. For outcome prediction, we included 206 patients: 144 scans were included in the training set, 21 in the validation set, and 41 in the test set. The classifiers that included the CAE and the radiomic features showed AUC values of 0.88 and 0.81, respectively, while the model based on FIV had an AUC of 0.79. This difference was not found to be statistically significant. Feature importance results showed that lesion intensity heterogeneity received more weight than lesion volume in outcome prediction. This study suggests that predictions of functional outcome should not be based on FIV alone and that FU-DWI images capture additional prognostic information.

## 1. Introduction

Acute ischemic stroke (AIS) has a major impact on patients’ lives: the majority of AIS patients do not return to functional independence or their functional status before experiencing AIS—even with adequate treatment [1]. Accurate estimations of functional outcome after treatment could help to guide patients in setting realistic expectations and deciding on the focus of the rehabilitation process [2].

Follow-up infarct volume (FIV) as measured by radiological follow-up imaging has been suggested as a prognostic marker for functional outcome [3]. However, previous studies have indicated that FIV is only moderately associated with functional outcome: only 12% of functional outcomes are explained by FIV [3]. It has been suggested that current imaging techniques—such as computed tomography (CT), CT perfusion, and diffusion-weighted imaging (DWI)—are not able to accurately predict or measure infarcted tissue [4]. A complicating factor is the fact that progression from severely ischemic tissue to actual infarction is likely not constant over time and not always clearly visible on CT or DWI scans. In addition, cells within the ischemic region may potentially remain viable, depending on their tolerance to ischemia [4]. A previous study showed that ischemic lesions may still evolve in the subacute phase even after successful treatment, resulting in smaller or larger lesions after 1-week follow-up [5]. 

Previous studies have suggested that tissue estimated as infarcted on radiological imaging may contain additional prognostic information that could improve outcome prediction for AIS. For example, intensity heterogeneity on images in infarcted regions may reflect a variance in tissue vulnerability to ischemia and may represent the degree of ischemia [6]. In addition, Wang et al. showed that textural features, including heterogeneity, assessed based on T2 FLAIR and ADC images were associated with follow-up NIHSS and modified Rankin Scale (mRS) scores [7]. Moreover, the shape of the lesion may contain important information on the potential progression from ischemia to infarcted tissue [8]. 

Previous studies have demonstrated the potential of machine learning (ML) to use automatically extracted imaging biomarkers for outcome prediction in AIS. For example, Qiu et al. [9] trained a support vector machine (SVM) to show that thrombus radiomic features were more predictive for recanalization in patients treated with intravenous alteplase than manually extracted thrombus features. In addition, Hilbert et al. [10] showed that features automatically extracted by an autoencoder combined with a dense layer outperformed ML models trained on handcrafted imaging biomarkers in predicting successful reperfusion and functional outcome at 90 days after stroke onset. 

We hypothesized that infarct volume alone, as measured by DWI, is not sufficient to represent the pathological changes in the ischemic brain region and that DWI data may contain additional prognostic information that is still unknown. We compared the performance of an ML model based on FIV alone with a radiomic-features-based model and a model based on features obtained from a deep learning autoencoder network in the prediction of favorable functional outcome.

## 2. Materials and Methods

Figure 1 shows the workflow of this study, which can be split up into two stages: feature extraction and outcome classification. Before we were able to extract features from the study dataset, we used an external dataset to train the CAE and a deep learning network for the delineation of the infarct lesions.

### 2.1. Datasets

The external dataset included patients from the HERMES collaboration [11] with available diffusion-weighted imaging (DWI) at 24 h; it also included DWI images from patients with subacute lesions from SISS ISLES 2015 [12]. The HERMES collaboration was formed to pool patient-level data from seven randomized, controlled clinical trials that showed the efficacy of endovascular treatment (EVT) over best medical management alone for patients with an occlusion of arteries of the proximal anterior circulation (ICA, M1, and M2) [11]. Each trial in the HERMES collaboration was approved by the relevant national or local medical ethical committee. All imaging data and clinical reports were anonymized, and informed consent was obtained for each patient according to each trial protocol. Patients included in these trials consented to participation in the individual trials as well as the use of their data for future research.

The study dataset included patients from the MR CLEAN-NO IV trial with available DWI at 24 h post-treatment. The MR CLEAN-NO IV trial was a randomized clinical trial in which the effect of immediate endovascular treatment (EVT) on 90-day functional outcome in patients with AIS was compared to that of intravenous treatment with alteplase (IVT) followed by endovascular treatment [13]. Patients who were directly admitted to an EVT-capable hospital were included if they were eligible for IVT and EVT and over the age of 18 with a proximal occlusion of the anterior circulation. Informed consent was obtained following a deferred consent procedure in accordance with national legislation in the three participating countries [14]. 

Since imaging was acquired in a multicenter and international setting, scanner types and image acquisition parameters varied. Images were acquired with a field strength of 1.5 or 3 Tesla. The slice thickness ranged from 3 to 6 mm. For this study, patients were excluded if DWI images contained motion artifacts or in cases with unsolvable registration errors.

### 2.2. Pre-Processing and Image Analysis

#### 2.2.1. Image Registration

All DWI images were transformed to standard MNI space via non-rigid registration using the SPM8 toolbox [15], resulting in isotropic voxel dimensions of 1 mm. Intensities were normalized using the white stripe normalization toolbox [16]. Images processed by the CAE were additionally subsampled to an isotropic voxel spacing of 3 mm before analysis.

#### 2.2.2. Lesion Segmentation

To delineate the infarct lesions for the MR CLEAN-NO IV population, we trained a Deepmedic network [17]. Deepmedic is a multi-scale 3D convolutional neural network with a fully connected conditional random field and has been shown to be computationally efficient; it performed best at brain lesion segmentation in the ISLES 2015 challenge [12]. We trained the network on the HERMES DWI images for which lesion segmentations were available [3]. Images were split into a training set (70%), a validation set (10%), and a test set (20%). The trained network was applied to the MR CLEAN-NO IV image dataset. Each resulting segmentation was checked by one of two experienced observers (authors J.W.H. and M.L.T.) and manually adjusted, in cases of erroneous segmentation, using ITK-SNAP [18]. Hemorrhagic transformation was included within the lesion. For cases where no consensus could be reached, the segmentations were assessed by two expert neuroradiologists (authors C.B.M. and M.S.K., with >20 and >5 years of experience, respectively) to reach a consensus.

### 2.3. Feature Extraction

#### 2.3.1. Convolutional Autoencoder

We developed and optimized a CAE for the reconstruction of DWI images (Figure 2) using the Keras libraries [19]. By learning how to reduce the dimensions of the feature space and reconstruct images from this low-dimensional feature space (latent space), the CAE learns the most important features that describe the source image. A CAE consists of several layers that downsample an image (encoder) to a compressed feature space (latent space), followed by several upsampling layers (decoder) that reverse the downsampling by upsampling the image to the original image dimensions. The encoder consisted of four 4 × 4 × 4 convolutional layers with stride 2 and rectified linear unit activation. Since each convolutional layer divides the feature space dimensions in half, it was favorable to use input dimensions that were powers of 2. Therefore, we first zero-padded the input image to the dimensions of 64 × 80 × 64. For each subsequent convolutional layer, the number of filters was doubled, starting at 16. Each convolutional layer was followed by group normalization to reduce the chance of overfitting. After the final convolutional layer of the encoder, the feature space was flattened, and a dense layer was added to reduce the number of features in the latent space to 100. The decoder reversed the encoder by first upsampling the feature space by a factor 2, followed by the use of a convolutional layer with stride 1 to maintain the feature space dimensions. The number of filters in the first four convolutional layers of the decoder was equal to the number in the encoder but in the opposite direction. Again, each convolutional layer was followed by group normalization. After the fourth convolutional layer, three additional convolutional layers were added to gradually reduce the fourth dimension of the feature space to 1, resulting in the input image dimensions. After the last convolutional layer, the output image was cropped to the original image dimensions. The loss function of the CAE was the mean squared error (MSE) between the source image and the resulting image.

For the development and optimization of the CAE, the data from the HERMES and ISLES challenges were combined and divided into a training (80%) set and a validation set (20%). To increase the number of training samples available, we performed data augmentation for the training set by flipping the images over the z-axis. The network was trained for 200 epochs with a batch size of 2. We used the validation set to optimize the CAE. After optimization, we extracted features from the MR CLEAN-NO IV DWI dataset.

#### 2.3.2. Radiomics

Radiomic features extracted from medical images aim to identify and quantify pathological effects that may be invisible to the human eye [20]. Radiomic features are extracted from a region of interest (ROI), in our case, the infarct lesion, and include first-order statistics, shape, and textural features (Figure 3). Examples of first-order statistics are minimum, maximum, and mean intensity within the lesion. Shape features contain both 2D metrics, such as the maximum diameter within a slice of the ROI, and 3D metrics, including the 3D volume of the ROI. Textural features are computed using filtering methods and matrices that capture the relationships between multiple voxels. An example of these matrices is the gray-level size zone matrix (GLSZM), which represents the number of neighboring pixels with the same intensity. Metrics computed from this matrix represent coarseness and homogeneity within the lesion. In total, 100 radiomic features were extracted using the PyRadiomic Toolkit [20].

### 2.4. Classification

An SVM classifier was optimized based on FIV, radiomic, and CAE features. The SVM classifier separated different outcome groups by optimizing a hyperplane that described the boundary with maximal distance between the features that belonged to the different outcome groups. We assessed its performance at accurately predicting functional independence, defined as an mRS score of 0–2, at 90 days. For the implementation of the SVM, we used the scikit-learn toolkits [21]. The optimization and testing were performed similarly for both feature sets. From the MR CLEAN-NO IV DWI dataset, 80% of the DWI images were used for 5-fold cross-validation. The remaining 20% of the images were used to test the performance of the final classifier. Before optimizing the SVM, all features were normalized with the scikit-learn ‘RobustScaler’ function, which scales each feature based on its median and interquartile ranges. To optimize the SVM, we performed a grid search to find the most optimal kernel type and coefficient (gamma) and regularization parameter (C). The following options were used: linear kernel type, radial basis function, polynomial or sigmoid, and gamma of 1 × 10^−2^ to 1 × 10^3^ per order of magnitude. The performance of the classifiers was evaluated based on the area under the receiver operating characteristic curve (AUC) computed for the test set. The AUCs were pairwise compared and tested for statistically significant differences, with the highest AUC as a reference, using deLong’s test [22]. Classification accuracy, precision, and recall were also reported. For the radiomics-based classifier, we investigated feature importance based on the Shapley additive explanation (SHAP) values [23]. For the CAE, we visualized a representative predicted validation image and compared it to the original validation image.

## 3. Results

### 3.1. Study Population

From the 307 patients with FU-DWI images in the HERMES dataset, we excluded 55 patients due to poor image quality. No images were excluded from the ISLES dataset (*n* = 64), which resulted in a total dataset of 316 images. From these images, 253 DWI images were included in the training set and 63 in the validation set. 

The MR CLEAN-NO IV dataset contained 220 patients with available FU-DWI scans. We excluded 11 patients due to poor image quality and 3 patients due to uncorrectable registration errors. This resulted in 206 patients in the study population, from which 144 scans were included in the training set, 21 in the validation set, and 41 in the test set. The baseline and follow-up characteristics for the MR CLEAN-NO IV subpopulation and the overall study population are provided in the Appendix A. 

### 3.2. Autoencoder Image Reconstruction

The training MSE of the CAE was 2.0 × 10^−3^ (arbitrary units), and the validation error was 5.1 × 10^−3^. Figure 4 shows the reconstruction of a validation image (left) established by the CAE (middle) and the corresponding difference map (right). The difference map shows small intensity differences in most of the healthy brain regions. The largest differences in intensities were present at the transition between brain tissue and cerebral spinal fluid. It is of note that some predicted voxels within the lesion and ventricles also differed in intensity from the original. The CAE was able to reconstruct the lesion at a location similar to that of the original image.

### 3.3. Functional Outcome Prediction

Table 1 shows the results of the best-performing classifiers that were trained on FIV, CAE-selected features, and radiomic features. We found the highest test accuracy for the FIV-based SVM classifier (0.74). The precision was highest for the radiomic-features-based SVM classifier (0.80), while the recall was highest for the SVM classifier based on FIV (0.73). Based on the AUC (Figure 5), the SVM classifier trained on radiomic features showed the best performance (0.88). However, this improved outcome prediction was not statistically significant compared to the model based on FIV (*p* = 0.15) or the model based on the CAE-trained SVM classifier (*p* = 0.37).

### 3.4. Radiomic Feature Importance

Figure 6 lists the 15 radiomic features with the largest impact on the outcome prediction generated by the SVM classifier based on SHAP values. The majority of these features consisted of textural features [24]. The two most important features were ‘large area of high gray-level emphasis’ and ‘large area of low gray-level emphasis’, which are both based on the GLSZM matrix. These features represented the presence of large areas with high or low intensities within the lesion and steered the classifier towards unfavorable functional outcome classifications. The lesion volume features with the most impact on the classification were mesh volume (volume based on the reconstructed 3D mesh based on the delineation) and voxel volume (lesion volume based on voxel volume). These features were in 9th and 10th place, respectively.

## 4. Discussion

We compared the predictive performance of ML models based on three different feature sets: FIV, radiomic, and CAE features. We showed that the accuracy of favorable outcome prediction based on radiological imaging characteristics was improved when using automatically extracted imaging biomarkers from FU-DWI images. However, we were unable to show statistically significant differences in independent data. We found that intensity heterogeneity in the FU-DWI lesion was most important for functional outcome prediction.

The model based on radiomic features most accurately predicted favorable functional outcome, and our SHAP analysis showed that its most important features were related to textural information. Thus, the SVM classifier weighted the decisions regarding predicted outcome mostly on texture and, to a lesser extent, on lesion volume. The most important textural features were related to intensity heterogeneity. This corresponds with the current literature [7] and may reflect the heterogeneity in tissue vulnerability. Possibly, intensity heterogeneity in our study population was related to the presence of hemorrhage, which is negatively associated with functional outcome [25]. We performed an explorative analysis to study whether hemorrhage was present in patients with heterogeneous lesions. Together with an expert neuroradiologist (C.B.M.), we visually inspected the DWI and T2* images of the patients with high values for heterogeneity and negative SHAP values (which corresponded to predictions of unfavorable outcome). Information about treatment outcome was not provided. Hemorrhage could not be observed in these patients. This study suggests that functional outcome predictions should not be based on FIV alone as an imaging biomarker and that FU-DWI images capture additional prognostic information about the ischemic tissue in patients with an LVO. 

The radiomic-feature-based SVM outperformed the CAE-feature-based SVM and was best at correctly classifying patients with favorable outcomes: 20% of the patients for whom a favorable outcome was predicted by the radiomic-features-based SVM did not achieve functional independence, compared to 25% for the CAE-feature-based SVM. However, the recall for the radiomic-feature-based classifier was only 65%, while for the CAE-feature-based SVM, 73% of patients with favorable outcomes were selected. Considering these results, we think that the CAE-feature-based SVM is more appropriate for clinical decision making since patients with the potential for a favorable outcome should not be missed. 

An advantage of the use of CAE features over radiomic features is that no lesion delineations are required for feature extraction. In this study, lesion delineation required manual annotations, which is time-consuming and introduces user dependency. In addition, since the CAE features are based on the entire brain volume, information on surrounding tissue relative to the lesion is incorporated. A disadvantage of using the CAE is that features are less interpretable. Future studies could potentially perform activation visualization to study which information about the brain was most important for the classifier [26]. In addition, ischemic lesion location could be a feature of interest for predicting functional outcome [27].

This study suffers from some limitations. First, selection bias might have occurred since, for functional outcome prediction, we only included patients who complied with the inclusion criteria of MR CLEAN-NO IV. Consequently, our results cannot be generalized to ischemic stroke patients with more distal occlusions, posterior circulation occlusions, or a stroke with minor symptoms. In addition, our results are not generalizable to patients not eligible for IVT and/or EVT, or who present outside the treatment window or with a baseline NIHSS < 2. In addition, in our healthcare system, a follow-up MRI for AIS patients is mostly only acquired in a research setting. Therefore, we only included patients from centers who participated in the MR CLEAN-NO IV trial in whom a follow-up MRI was performed as a secondary outcome measure as required by the trial protocol. Second, the mRS score was used since it is a common endpoint in AIS trials for the assessment of independence in daily activities. However, it is coarse and mainly focuses on motor function, with less attention to the assessment of cognitive function and emotional processing. Third, the performance of the CAE may have been hampered due to the optimization process of the CAE itself. Improving the CAE, for example, by adding more data to the training set, may improve the accuracy of classifications of functional independence by the SVM based on CAE features.

## Figures and Tables

**Figure 1 diagnostics-12-01786-f001:**
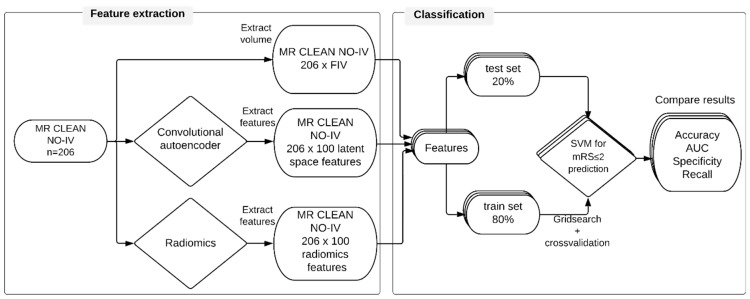
Study workflow for functional outcome prediction. Three different feature sets were extracted: follow-up infarct volume, features extracted by a convolutional autoencoder, and radiomic features. Each feature set was split into a training (80%) set and a test (20%) set. A support vector machine (SVM) was trained on the training set to classify favorable outcome. The SVMs were tested on the test set. The results were evaluated for each SVM.

**Figure 2 diagnostics-12-01786-f002:**
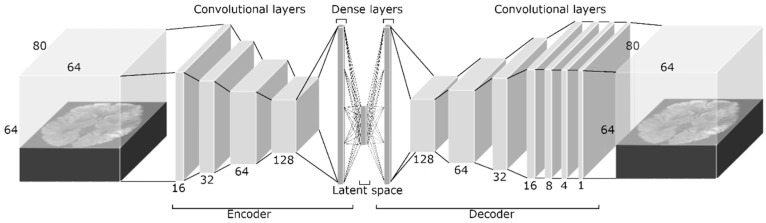
The convolutional autoencoder architecture. The dimensions of the input image were 64 × 80 × 64. The encoder consisted of four 4 × 4 × 4 convolutional layers with stride 2 and rectified linear unit activation. For each subsequent convolutional layer, the number of filters was doubled, starting at 16. Each convolutional layer was followed by group normalization. After the final convolutional layer of the encoder, the feature space was flattened, and a dense layer was added. The decoder contained the same components as the encoder in the opposite direction, except that the feature space was upsampled first by a factor of 2, and the stride of the convolutional layers was kept at 1. After the fourth convolutional layer, three additional convolutional layers reduced the fourth dimension of feature space to 1, resulting in the original image dimensions.

**Figure 3 diagnostics-12-01786-f003:**
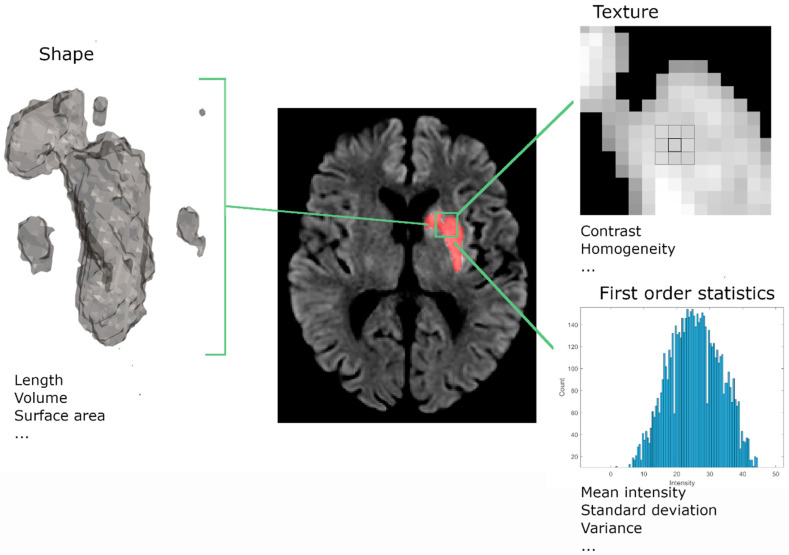
Illustration of the three radiomic feature classes. Radiomic features consist of shape, texture, and first-order statistics features. Shape features describe the 2D and 3D size and shape of the lesion. Textural features describe the intensity distribution and relations between neighboring voxels. First-order statistics describe the intensity distributions of the lesion.

**Figure 4 diagnostics-12-01786-f004:**
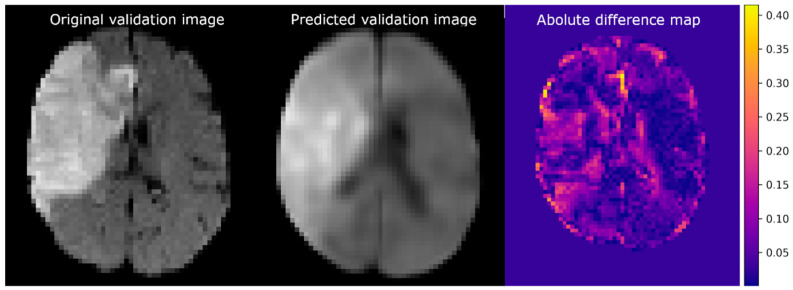
Example of imaging reconstruction using a trained convolutional autoencoder. (**Left**) An axial slice of the original validation image. (**Middle**) The corresponding slice of the predicted image. (**Right**) The absolute difference between the normalized original and predicted images.

**Figure 5 diagnostics-12-01786-f005:**
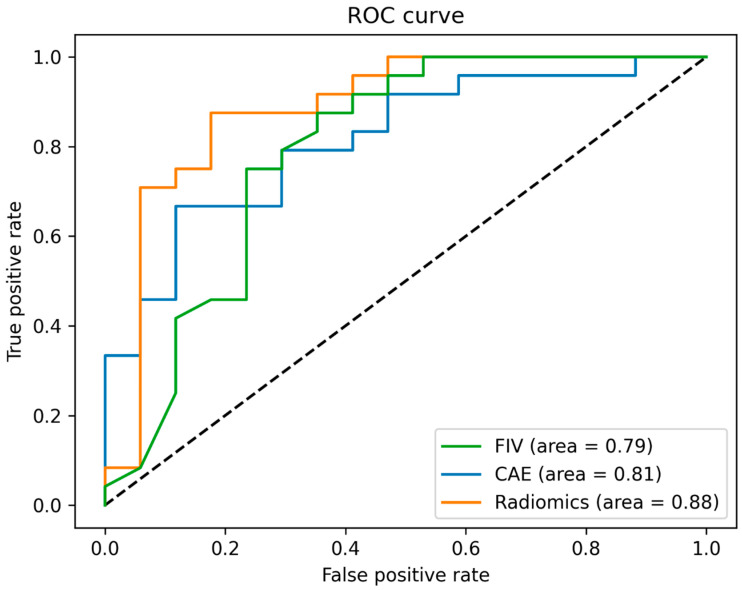
Receiver operating curves for the best-performing support vector machine model based on three different inputs: features extracted by a convolutional autoencoder, radiomic features, and follow-up infarct volume.

**Figure 6 diagnostics-12-01786-f006:**
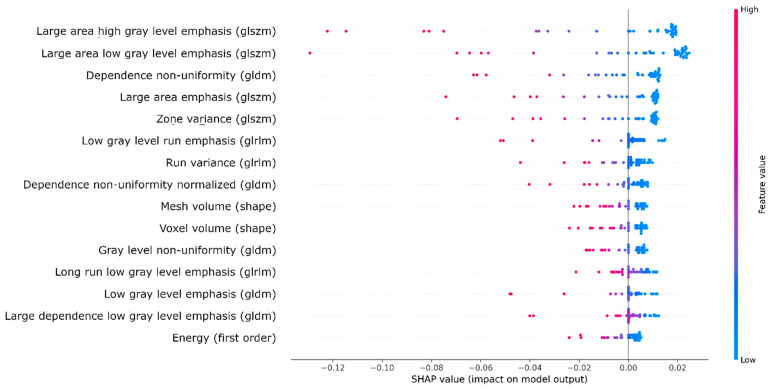
SHAP summary plot showing the top 15 radiomic features (and their feature classes) in terms of impact on the classification based on the SHAP values. Negative and positive SHAP values represent unfavorable and favorable outcome classifications, respectively. The feature values are represented by a color map, ranging from blue (low value) to red (high value). Abbreviations of second-order radiomic feature classes in gray-level matrices: size zone (glszm), dependence (gldm), and run length (glrlm).

**Table 1 diagnostics-12-01786-t001:** Training and testing accuracy, AUC, precision, and recall for the best-performing SVM classifiers based on FIV, the autoencoder features, and the radiomic features. The *p*-values resulting from deLong’s tests against the radiomic features are presented in the last column.

Feature Extraction Method	Training Accuracy (*n* = 144)	Testing Accuracy (*n* = 41)	AUC (*n* = 41)	Precision (*n* = 41)	Recall (*n* = 41)	deLong’s Test *p*-Value
FIV only *	0.73	0.74	0.79	0.78	0.73	0.15
Autoencoder **	0.76	0.71	0.81	0.70	0.71	0.37
Radiomics ***	0.75	0.71	0.88	0.80	0.65	

* SVM parameters: {C: 1000, gamma: 0.01, kernel: rbf}, ** SVM parameters: {C: 0.1, gamma: 0.01, kernel: linear}, *** SVM parameters: {C: 1, gamma: 0.001, kernel: sigmoid}.

## Data Availability

The SISS ISLES 2015 is an open dataset that is available after registration with https://www.smir.ch (accessed on 17 July 2022). Trial data can be made available on reasonable request via mrclean@erasmusmc.nl.

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
