# Peer review of "Outcome Prediction Based on Automatically Extracted Infarct Core Image Features in Patients with Acute Ischemic Stroke"

_diagnostics, 2022, doi:10.3390/diagnostics12081786_

Round 1
Reviewer 1 Report
Scineetifically correct paper with absolutely interesting contents
Author Response
“Scientifically correct paper with absolutely interesting contents”
We thank the reviewer for the feedback on our manuscript entitled: “Outcome prediction based on automatically extracted infarct core image features in patients with acute ischemic stroke”. We are very pleased to hear that the reviewer thought that our study was scientifically correct and interesting. The reviewer has noted that the paper requires moderate English changes. We have inspected the text for grammar/textual mistakes and made corrections where needed.
Reviewer 2 Report
In this study the authors aimed to evaluate how follow-up diffusion-weighted imaging (FU-DWI) could contain other imaging biomarkers which could aid in improving outcome prediction models for acute ischemic stroke. They tested three SVM models over 206 patients and found that lesion intensity heterogeneity received more weight than lesion volume in outcome prediction by 82 the classifier.
Statistical analyses have well conducted and figures and tables are informative and clear.
The paper is interesting, finely written and worthy of publication, I have no concern.
Author Response
In this study the authors aimed to evaluate how follow-up diffusion-weighted imaging (FU-DWI) could contain other imaging biomarkers which could aid in improving outcome prediction models for acute ischemic stroke. They tested three SVM models over 206 patients and found that lesion intensity heterogeneity received more weight than lesion volume in outcome prediction by 82 the classifier.
Statistical analyses have well conducted and figures and tables are informative and clear.
The paper is interesting, finely written and worthy of publication, I have no concern.
We thank the reviewer for the feedback on our manuscript. We are very pleased to hear that the reviewer has no concerns.
The reviewer has noted that the paper requires spell checks. We have inspected the text for grammar/textual mistakes and made corrections to the minor spelling mistakes.
Reviewer 3 Report
The authors present a highly relevant study on machine learning strategies to detect imaging features that are prognostically impact outcome in stroke patients. I have only minor concerns that need to be adressed:
In the Abstract, SMV needs to be explained. There is a lack of conclusion and explanation of the clinical relevance of the findings, what was the significant finding in predicting outcome in this study?
The authors state, that the most important textural features relate to intensity heterogeneity wich might reflect the heterogeneity in tissue vulnerability. The postulated that intensity heterogeneity in their study population relates to the presence of hemorrhage, which is negatively associated with functional outcome. They performed visual inspection of the results, and did not find a relation between the textural features and the presence of hemorrhage. How did the authors account for the bias in a visual assessment? Was this performed by a single person? Was this person blinded for outcome and dataset?
Author Response
We thank the reviewer for the feedback on our manuscript We are very pleased to hear that the reviewer thought that our study was highly relevant. We have carefully reviewed the comments and revised the manuscript accordingly. A point-by-point response to the remarks is given below:
The authors present a highly relevant study on machine learning strategies to detect imaging features that are prognostically impact outcome in stroke patients. I have only minor concerns that need to be addressed:
In the Abstract, SMV needs to be explained. There is a lack of conclusion and explanation of the clinical relevance of the findings, what was the significant finding in predicting outcome in this study?
We thank the reviewer for pointing out that the abstract missed an explanation of SVM. We noticed that the terms used for our machine learning models in the abstract were inconsistent in level of detail. We have adjusted the terms to more generally known terms:
“Lesions were segmented using a deep learning model trained on the HERMES and ISLES datasets. We assessed the performance of three classifiers in predicting functional independence for the MR CLEAN-NO IV trial cohort, based on: (1) FIV alone, (2) the most important features obtained from a trained convolutional autoencoder (CAE), and (3) radiomics.”
We hope that the reviewer agrees that, considering the limited number of words that are allowed for the abstract, it is very challenging to include a clear explanation of SVM without hampering an appropriate description of our study because other elaborations need to be reduced. However, we also noticed that no explanation was given of SVM within the main text. We have therefore added it within the methods section:
“The SVM classifier separates different outcome groups by optimizing a hyperplane that describes the boundary with maximal distance between the features that belong to the different outcome groups.”
We also agree with the reviewer that the abstract lacked a conclusion and explanation of the clinical relevance of the findings. We have improved the conclusion within the abstract accordingly:
“Feature importance results showed that lesion intensity heterogeneity received more weight than lesion volume in outcome prediction. This study suggests that functional outcome should not be based on FIV alone and that FU-DWI images capture additional prognostic information.”
The authors state, that the most important textural features relate to intensity heterogeneity which might reflect the heterogeneity in tissue vulnerability. The postulated that intensity heterogeneity in their study population relates to the presence of hemorrhage, which is negatively associated with functional outcome. They performed visual inspection of the results, and did not find a relation between the textural features and the presence of hemorrhage. How did the authors account for the bias in a visual assessment? Was this performed by a single person? Was this person blinded for outcome and dataset?
We thank the reviewers for the question about the way we approached our visual inspection. In order to minimize the bias in visual assessment, we consulted an expert neuroradiologist who was not aware of the patient’s outcome. The visual inspection was not performed by multiple assessors, since it was an explorative analysis. We have improved our explanation of the followed procedure:
"Possibly, intensity heterogeneity in our study population relates to the presence of hemorrhage, which is negatively associated with functional outcome[25]. We performed an explorative analysis to study if hemorrhage was present within patients with heterogeneous lesions. Together with an expert neuroradiologist (C.B.L.M.) we visually inspected the DWI and T2* images of the patients with high values for heterogeneity and negative SHAP values (which corresponds to the prediction of unfavorable outcome). Information about treatment outcome was not provided. Hemorrhage could not be observed in these patients."
The reviewer has noted that the paper requires spell checks. We have inspected the text for grammar/textual mistakes and made corrections to the minor spelling mistakes.